# Explanation-Based Attention for Semi-Supervised Deep Active Learning

**Denis Gudovskiy, Alec Hodgkinson**
Panasonic Beta Research Lab, Mountain View, CA, 94043, USA
{denis.gudovskiy, alec.hodgkinson}@us.panasonic.com

**Takuya Yamaguchi, Sotaro Tsukizawa**
Panasonic AI Solutions Center, Osaka, Japan
{yamaguchi.takuya2015, tsukizawa.sotaro}@jp.panasonic.com

## Abstract

We introduce an attention mechanism to improve feature extraction for deep active learning (AL) in the semi-supervised setting. The proposed attention mechanism is based on recent methods to visually explain predictions made by DNNs. We apply the proposed explanation-based attention to MNIST and SVHN classification. The conducted experiments show accuracy improvements for the original and class-imbalanced datasets with the same number of training examples and faster long-tail convergence compared to uncertainty-based methods.

## 1 Introduction

Deep active learning (AL) minimizes the number of expensive annotations needed to train DNNs by selecting a subset of relevant data points from a large unlabeled dataset (Lewis & Gale, 1994). This subset is annotated and added to the training dataset in a single *pool* of data points or, more often, in an iterative fashion. The goal is to maximize prediction accuracy while minimizing the product of pool size $\times$ number of iterations. A proxy for this goal could be the task of matching feature distributions between the validation and the AL-selected training datasets.

In density-based AL approaches, data selection is typically performed using a simple $L_2$-distance metric (Sener & Savarese, 2018). The image retrieval field (Zhou et al., 2017) has advanced much further in this area. For example, recent state-of-the-art image retrieval systems are based on DNN-based feature extraction (Babenko & Lempitsky, 2015) with attention mechanisms (Noh et al., 2017). The latter estimates an attention mask to weight importance of the extracted features and it is trained along with the feature extraction.

Inspired by this, we employ image retrieval techniques and propose a novel attention mechanism for deep AL. Unlike supervised self-attention in (Noh et al., 2017; Vaswani et al., 2017), our attention mechanism is not trained with the model. It relies on recent methods to generate visual explanations and to attribute feature importance values (Sundararajan et al., 2017). We show the effectiveness of such explanation-based attention (EBA) mechanism for AL when combined with multi-scale feature extraction on a number of image classification datasets. We also conduct experiments for distorted class-imbalanced training data which is a more realistic assumption for unlabeled data.

## 2 Related Work

AL is a well-studied approach to decrease annotation costs in a traditional machine learning pipelines (Settles, 2010). Recently, AL has been applied to data-demanding DNN-based systems in semi-supervised or weakly-supervised settings. Though AL is an attractive direction, existing methods struggle to deal with high-dimensional data e.g. images. We believe this is related to the lack of class and instance-level feature importance information as well as the inability to capture spatially-localized features. To overcome these limitations, we are interested in estimating spatially-multiscale features and using our EBA mechanism to select only the most discriminative features.

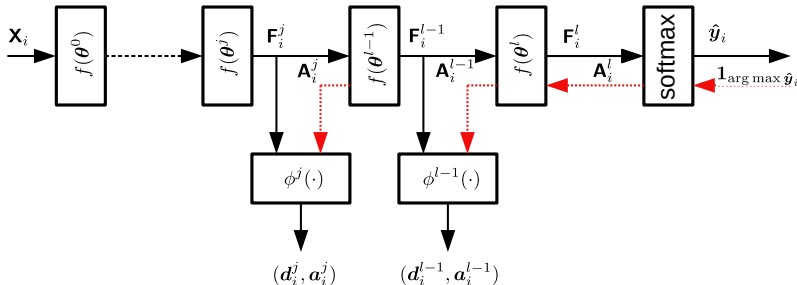

Figure 1: Conventional multi-scale feature extraction and the proposed EBA extension (dashed).

Wang et al. (2017) proposed to augment the training dataset by labeling the least confident data points and heuristically pseudo-labeling high confidence predictions. We believe the softmax output is not a reliable proxy for the goals of AL i.e. for selecting images using feature distribution matching between validation and train data. Unlike (Wang et al., 2017), we use pseudo labels only to estimate EBA vectors and find similarities between discriminative features.

Gal et al. (2017) introduced a measure of uncertainty for approximate Bayesian inference that can be estimated using stochastic forward passes through a DNN with dropout layers. An acquisition function then selects data points with the highest uncertainty which is measured at the output of softmax using several metrics. Recent work (Beluch et al., 2018) extended this method by using an ensemble of networks for uncertainty estimation and achieved superior accuracy.

Sener & Savarese (2018) formulated feature similarity-based selection as a geometric core-set approach which outperforms greedy $k$-center clustering. Though their method can complement our approach, we are focusing on the novel feature extraction. For instance, they employed a simple $L_2$ distance similarity measure for the activations of the last fully-connected layer.

The most similar work to ours, by Vodrahalli et al. (2018), uses the gradients as a measure of importance for dataset subsampling and analysis. However, our approach formulates the problem as a multi-scale EBA for AL application and goes beyond a less robust single-step gradient attention. Other related works are online importance sampling methods (Ren et al., 2018) and the influence functions approach in (Koh & Liang, 2017). Online importance sampling upweights samples within the mini-batch during supervised training using gradient similarity while influence functions analyze data point importance using computationally challenging second-order gradient information.

## 3 METHOD

**Pool-based AL.** Let $(\mathbf{X}, y)$ be an input-label pair. There is a validation dataset $\{(\mathbf{X}_i^v, y_i^v)\}_{i \in \mathbb{M}}$ of size $M$ and a collection of training pairs $\{(\mathbf{X}_i, y_i)\}_{i \in \mathbb{N}}$ of size $N$ for which, initially, only a small random subset or *pool* of labels indexed by $\mathbb{N}^1$ is known. The validation dataset approximates the distribution of test data. At every $b$th iteration the AL algorithm selects a pool of $P$ new labels to be annotated and added to existing training pairs which creates a training dataset indexed by $\mathbb{N}^b$.

A DNN $\Phi(\mathbf{X}, \boldsymbol{\theta})$ is optimized by minimizing a loss function $(N^b)^{-1} \sum_{i \in \mathbb{N}^b} L(\hat{\boldsymbol{y}}_i, y_i)$ w.r.t. to model parameters $\boldsymbol{\theta}$. However, the actual task is to minimize validation loss expressed by $M^{-1} \sum_{i \in \mathbb{M}} L(\hat{\boldsymbol{y}}_i^v, y_i^v)$. Therefore, an oracle AL algorithm achieves minimum of *validation loss* using the smallest $b \times P$ product. In this work, we are interested not in finding an oracle acquisition function, but in a method to extract relevant features for such function. We use a low-complexity greedy $k$-center algorithm to select the data points in the unlabeled training collection which are most similar to the misclassified entries in the validation dataset.

**Feature descriptors.** Let $\mathbf{F}_i^j \in \mathbb{R}^{C \times H \times W}$, where $C$, $H$, and $W$ are the number of channels, the height, and the width, respectively be the output of the $j$th layer of DNN for input image $\mathbf{X}_i$. Then, a feature vector or *descriptor* of length $L$ can be defined as $\boldsymbol{d}_i = \phi(\mathbf{F}_i) \in \mathbb{R}^{L \times 1}$, where function $\phi(\cdot)$ is a conventional average pooling operation from (Babenko & Lempitsky, 2015). In a multi-scale case, descriptor is a concatenation of multiple feature vectors $\boldsymbol{d}_i = [\phi^j(\mathbf{F}_i^j), \cdots, \phi^l(\mathbf{F}_i^l)]$.

A descriptor matrix for the validation dataset $\boldsymbol{V}_d \in \mathbb{R}^{L \times M}$ and training dataset $\boldsymbol{S}_d \in \mathbb{R}^{L \times N}$ can be calculated using forward passes. Practically, descriptors can be compressed for storage efficiency reasons using PCA, quantization, etc. Then, a match kernel (Lee, 1999), e.g. cosine similarity, can be used to match features in both datasets. Assuming that vectors $\boldsymbol{d}_i$ are $L_2$-normalized, the cosine-similarity matrix is simply $\boldsymbol{R}_d = \boldsymbol{V}_d^T \boldsymbol{S}_d$.

**Explanation-based attention.** Feature maps $\mathsf{F}_i$ extracted by $\Phi(\mathbf{X}, \boldsymbol{\theta})$ and pooled by $\phi(\cdot)$ contain features that: a) are not class and instance-level discriminative (in other words, not disentangled), b) spatially represent features for a plurality of objects in the input. We would like to upweight discriminative features that satisfy a) and b) using an attention mechanism. One approach would be to use self-attention (Vaswani et al., 2017) at the cost of modifying network architecture and intervening into the training process. Instead, we propose to use EBA that is generated only for feature selection. The EBA mechanism attributes feature importance values w.r.t. to the output predictions. Unlike a *visual explanation* task, which estimates importance heatmaps in the input (image) space, we propose to estimate feature importance tensors $\mathsf{A}_i$ of the internal DNN representations $\mathsf{F}_i$. Attention tensors $\mathsf{A}_i$ can be efficiently calculated using a series of backpropagation passes. Using one of backpropagation-based methods called *integrated gradients* (IG) from (Sundararajan et al., 2017), $\mathsf{A}_i^j$ can be estimated as

$$\mathsf{A}_i^j = \frac{1}{K} \sum_{k=1}^{K} \frac{\partial L(\hat{\boldsymbol{y}}_i(k), y_i)}{\partial \mathsf{F}_i^j} = \frac{1}{K} \sum_{k=1}^{K} \frac{\partial L(\Phi(k\mathbf{X}_i/K, \boldsymbol{\theta}), y_i)}{\partial \mathsf{F}_i^j}, \tag{1}$$

where $K$ is the number of steps to approximate the continuous integral by a linear path. Other forms of (1) are possible: from the simplest *saliency* method for which $K = 1$ (Simonyan et al., 2014) to more advanced methods with randomly sampled input features (Gudovskiy et al., 2018).

Due to lack of labels $y_i$ in (1), we use common pseudo-labeling strategy: $y_i = \mathbf{1}_{\arg\max \hat{\boldsymbol{y}}_i}$. It is schematically shown in Figure 1. Unlike (Wang et al., 2017), pseudo-labels are used only to calculate similarity without additional hyperparameters rather than to perform a threshold-selected greedy augmentation. The EBA $\mathsf{A}_i$ can be converted to multi-scale attention vector using the same processing $\boldsymbol{a}_i = \phi(\mathsf{A}_i) \in \mathbb{R}^{L \times 1}$, which, by analogy, forms validation $\boldsymbol{V}_a \in \mathbb{R}^{L \times M}$ and train attention matrices $\boldsymbol{S}_a \in \mathbb{R}^{L \times N}$. The latter processing is implemented in most modern frameworks and, therefore, the complexity to generate $\mathsf{A}_i$ is only $K$ forward-backward passes.

**Summary for the proposed method.** A random subset of $N^1$ training data points is annotated and a DNN $\Phi(\mathbf{X}, \boldsymbol{\theta})$ optimized for this subset. Then, the AL algorithm iteratively ($b = 2, 3 \ldots$) performs following steps: 1) generates descriptor-attention matrix pairs $(\boldsymbol{V}_d, \boldsymbol{V}_a), (\boldsymbol{S}_d, \boldsymbol{S}_a)$, 2) calculates similarity matrix $\boldsymbol{R} = \boldsymbol{R}_d \odot \boldsymbol{R}_a = (\boldsymbol{V}_d^T \boldsymbol{S}_d) \odot (\boldsymbol{V}_a^T \boldsymbol{S}_a)$, where $\odot$ is element-wise product, 3) selects $P$ relevant data points from the remaining subset using acquisition function $\arg\max_{i \in \mathbb{N} \backslash \mathbb{N}^{b-1}}(\boldsymbol{R}(\mathbf{X}_i), \Phi)$ and 4) retrains $\Phi(\mathbf{X}, \boldsymbol{\theta})$ using augmented subset $\mathbb{N}^b$.

# 4 EXPERIMENTS

Our method as well as uncertainty-based methods from (Gal et al., 2017) are applied to the MNIST and SVHN classification. We evaluate AL with the original and distorted training data because unlabeled collection of data points cannot be a-priori perfectly selected. Hence, we introduce a *class imbalance* which is defined as the ratio of $\{0 \ldots 4\}$ to $\{5 \ldots 9\}$ digits. The following methods have been employed: *random* sampling, uncertainty-based (uncert), *greedy* selection using similarity matching without (top-P:none) and with EBA. The latter is estimated by saliency (top-P:grad) or IG (top-P:ig). We rerun experiments 10 times for MNIST and 5 times for SVHN with all-randomized initial parameters. Mean accuracy and standard deviation are reported. DNN parameters are trained from scratch initially and after each AL iteration. Mini-batch size is chosen by cross-validation.

**MNIST.** The dataset train/val/test split is 50K/10K/10K. The LeNet is used with the following hyperparameters: epochs=50, batch-size=25, lr=0.05, lr-decay=0.1 every 15 epochs, *uncert* methods and IG EBA use $K = 128$ passes and $L$ is 20 for single-scale (before *fc1* layer) and 90 for multi-scale descriptors (all layers are concatenated). Figure 2(a) shows that feature-only matching (top-P:none_L20) outperforms random selection by $\approx 1\%$ while EBA (top-P:ig_L90) adds another $1\%$ of accuracy when there is no class imbalance. High class imbalance (Figure 2(c)) increases that gap: up to 20% for feature-only matching and 25% with EBA. The highest accuracy is achieved by multi-

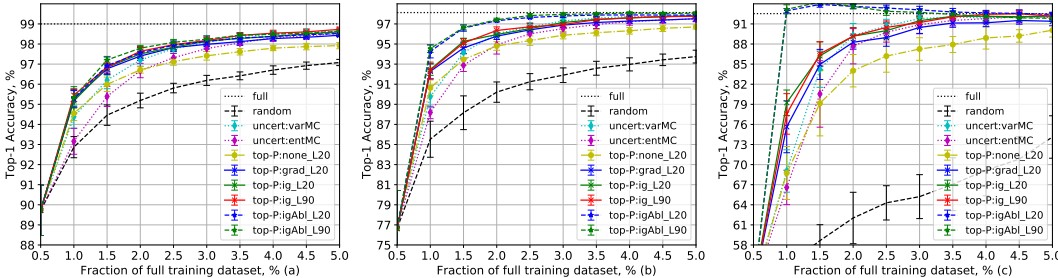

Figure 2: MNIST test dataset accuracy for 3 class imbalance ratios: a) 1 (no imbalance), b) 10 and c) 100. Total 9 AL iterations ($b = 10$) are performed each with $P = 250$ pool size.

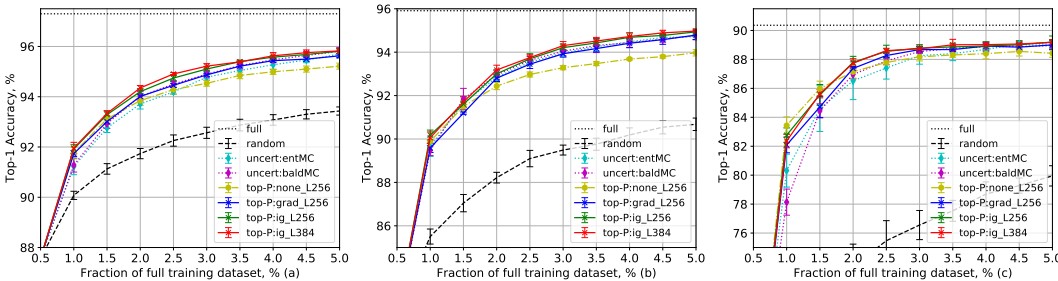

Figure 3: SVHN test dataset accuracy for 3 class imbalance ratios: a) 1 (no imbalance), b) 10 and c) 100. Total 9 AL iterations ($b = 10$) are performed each with $P = 2,500$ pool size.

scale EBA estimated by IG. EBA-based methods outperform the best uncertainty-based *variation ratio* (uncert:varMC) approach for all class imbalance settings except the last one where its accuracy is higher by less than 1% when $b = 4$. This might be related to small-scale MNIST and pseudo-label noise for EBA. To study the effects of pseudo-labeling, we plot true-label configurations (marked by "Abl") as well. The accuracy gap between EBA using true- and pseudo-labels is small with no class imbalance, but much larger (up to 25%) when class imbalance ratio is 100 during first AL iterations.

**SVHN.** The dataset train/validation/test split is 500K/104K/26K. A typical 8-layer CNN is used with the following hyperparameters: epochs=35, batch-size=25, lr=0.1, lr-decay=0.1 every 15 epochs, *uncert* methods and IG EBA use $K = 128$ and $L$ is 256 for single-scale (before *fc1* layer) and 384 for two-scale descriptors (+ layer before *conv7*). Figure 3 shows that the gap between random selection and the best EBA-based AL method grows from 2% to more than 12% when the unlabeled training collection has more class imbalance. The gap between full training dataset accuracy increases for larger-scale SVHN as well. This results in even faster convergence for the proposed AL relative to random selection. Accuracies of the *uncert* methods are closer to each other than for MNIST, which may signal their declining effectiveness for large-scale data. The proposed EBA-based methods outperform all uncertainty-based methods for SVHN in the first AL iterations (up to +2.5%) and later arrive at approximately equal results.

## 5 CONCLUSIONS AND FUTURE WORK

We applied recent image retrieval feature-extraction techniques to deep AL and introduced a novel EBA mechanism to improve feature-similarity matching. First feasibility experiments on MNIST and SVHN datasets showed advantages of EBA to improve density-based AL. Rather than performing AL for the well-picked training datasets, we also considered more realistic and challenging scenarios with class-imbalanced training collections where the proposed method emphasized the importance of additional feature supervision. In future research, EBA could be evaluated with other types of data distortions and biases: within-class bias, adversarial examples, etc. Furthermore, such applications as object detection and image segmentation may benefit more from EBA because multi-scale attention can focus on spatially-important features.

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
