# OpenReview forum: "Explanation-Based Attention for Semi-Supervised Deep Active Learning"
_ICLR.cc/2019/Workshop/LLD — LLD 2019_

### Official Review · AnonReviewer1 · 2019-04-10
**A small but focused contribution on active learning**

**Rating:** 4
**Confidence:** 2

**Review:**

This paper presents a novel method to match feature similarities for selecting unlabeled samples for active learning to train a model more label-efficiently.
The major innovation is to utilize Explanation-Based Attention (EBA) mechanism to improve matching feature similarities, which has been proved effective to attribute feature importance in computer vision domains.
The experiments show it outperforms conventional uncertainty-based approaches, especially when classes are imbalanced.

Overall, this paper is a small but focused contribution on active learning and well-written, clear for readers to follow.
The presentation can be improved with a more detailed description of notations (e,g. N_b and V_a are not explained, though it's easy to guess their meanings).
An illustrative figures of workflow mentioned in the section "summary for the proposed method" would be a plus.
The paper also enjoys the merit that it has a brief, clear overview of recent AL research to put itself in a broader context.

My major concerns are two-fold:
1) the intuition of utilizing integrated gradients (IG) and pseudo-labels is not super clear to me; 2) experiments should be more extensive.
The authors assume that the way of using IGs as EBA for evaluating sample similarity by multiplying themselves with descriptor matrices can upweight features that "a) are not class and instance-level discriminative, b) spatially represent features for a plurality of objects in the input. "
The assumption needs more justification.
For example, a) why to use average pooling function for both gradients and features is reasonable, b) the derivation of R_b is of what properties such that the distribution between training data and validation set are more similar iteratively (so we can believe the set of b-th iteration is better than the set of {b-1}-th). Also, experiments can justify the assumption as well with more visual explanations on why the proposed AL method is better and reasonable.

For the title, I suggest the authors not to use "explanation-based" since it is a little bit misleading. Readers may expect the authors use some kinds of explanitions to improve AL. I would say "Integrated Gradients-based Attention for Deep Active Learning" would be better.

That being said, I enjoyed reading this paper and would like to see it accepted with better presentation and more justification, experiments.

---

### Official Review · AnonReviewer2 · 2019-04-14
**Review of "Explanation-Based Attention for Semi-Supervised Deep Active Learning"**

**Rating:** 4
**Confidence:** 1

**Review:**

The authors consider the setting of deep attention learning. It consists in selecting critical unlabelled data in a semi-labelled dataset, so that once labelled they can improve drastically the accuracy of the model. The approach of the authors consist in training a DNN that computes similarity between data, starting with a limited pool of labelled data points. To do so, they augment iteratively the dataset using a greedy approach.

The paper is well written, and even I am not not at all a specialist of the field I think I understood the main points of the paper.

The numerical experiments seems strong enough to be convinced by their approach.

---

### Decision · Program_Chairs · 2019-04-16
**Acceptance Decision**

Accept